# CAN KNOWLEDGE GRAPHS MAKE LARGE LANGUAGE MODELS MORE TRUSTWORTHY? AN EMPIRICAL STUDY OVER OPEN-ENDED QUESTION ANSWERING

**Yuan Sui**[1], **Yufei He**[1], **Zifeng Ding**[2], **Bryan Hooi**[1]
[1] National University of Singapore [2] University of Cambridge
{yuansui, yufei.he, bhooi}@comp.nus.edu.sg, zd320@cam.ac.uk

## ABSTRACT

Recent works integrating Knowledge Graphs (KGs) have led to promising improvements in enhancing the reasoning accuracy of Large Language Models (LLMs). However, current benchmarks focus mainly on closed-ended tasks, leaving a gap in the assessment of more complex real-world scenarios. This gap has also obscured the evaluation of KGs' potential to mitigate the problem of hallucination in LLMs. To fill the gap, we introduce OKGQA, a new benchmark specifically designed to assess LLMs enhanced with KGs under open-ended, real-world question answering scenarios. OKGQA is designed to closely reflect the complexities of practical applications using questions from different types, and incorporates specific metrics to measure both hallucination ratio and the enhancement in reasoning capabilities. To consider the scenario in which KGs may have varying levels of mistakes, we propose another benchmark variant OKGQA-P to assess model performance when the semantics and structure of KGs are deliberately perturbed and contaminated. OKGQA aims to (1) explore whether KGs can make LLMs more trustworthy in an open-ended setting, and (2) conduct a comparative analysis to shed light on method design. We believe that this study can facilitate a more complete performance comparison and encourage continuous improvement in integrating KGs with LLMs to reduce hallucination.

## 1 INTRODUCTION

Contemporary LLMs are prone to producing **hallucinations** due to gaps in their knowledge (Gekhman et al., 2024; Lee et al., 2023). These inaccuracies commonly stem from misinformation, biases, or errors in the training data, and lead to responses that seem plausible but may be irrelevant or incorrect (Weng, 2024). This issue is particularly concerning in high-stakes contexts such as healthcare (He et al., 2023) and science (Taylor et al., 2022)[1].

To address this limitation, researchers have turned to leveraging external knowledge graphs (KGs) as a complementary (Pan et al., 2024; Luo et al., 2023a; Hu et al., 2023; Sui et al., 2024). KGs offer structured and explicit factual information—often domain-specific—and allow each piece of data to be traced back to its source (Zheng et al., 2023; Agrawal et al., 2023). This traceability not only enables verification of the model's reasoning but also brings transparency to the decision-making process, making KGs a promising method for enhancing the reliability of LLM reasoning. Find more details of related works in §D.

However, current benchmarks for testing these LLM+KG models are predominantly **closed-ended** (Jin et al., 2020; Puerto et al., 2023), restricting the model's output to a limited set of entities, relations (Talmor et al., 2019; Mihaylov et al., 2018) or logical forms (Yih et al., 2016; Talmor & Berant, 2018). While these benchmarks are useful to measure retrieval and basic reasoning, they do not adequately capture whether a model is **hallucinating**. In closed-ended settings, errors can stem from incorrect retrieval or from fabricating (hallucinating) answers, yet conventional metrics (e.g., accuracy or precision) cannot distinguish between these two issues. This becomes problematic for more complex, real-world applications that demand nuanced answers (Kantharaj et al., 2022).

---

[1]Code and data are released at https://anonymous.4open.science/r/OKGQA-CBB0

In contrast, our work focuses on open-ended KGQA, where LLMs are prompted to generate more elaborate answers, include reasoning paths and supporting facts derived from the KG (as shown in Figure 1). This broader output space offers two key advantages: First, it enables direct assessments of hallucination with metrics like FActScore (Min et al., 2023) or SAFE (Wei et al., 2024)), which decompose longer responses into atomic statements for factual consistency checks with external knowledge

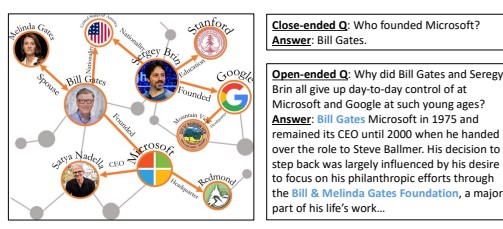

Figure 1: Comparison between Open-ended Question and Close-ended Question over Knowledge Graph.

sources like Wikipedia. Second, it increases the likelihood of exposing factual errors which helps to assess the phenomenon indicated in Qiu et al. (2024): where longer, more complex responses provide more opportunities for errors to occur. By adopting this open-ended approach, we aim to (1) explore whether KGs can make LLMs more trustworthy in the open-ended setting, and (2) conduct a comparative analysis to shed light on methods design and direction for leveraging KGs to reduce LLMs' hallucination.

To achieve this, we introduce a new benchmark, **O**pen-ended **K**nowledge-**G**raphs **Q**uestion **A**nswering (OKGQA), specifically designed to assess LLMs enhanced with KGs in an open-ended QA setting. OKGQA is designed to closely reflect the complexities of practical applications with diverse questions types as mentioned in Table 1, ensuring that all the queries cannot be answered simply by retrieving isolated KG facts. To consider the scenarios for potentially **contaminated** or **imperfect** KGs (*i.e.*, attributes may be mislabeled, relations may not exist, etc.), we also propose a variant OKGQA-P (§3.2) to assess model performance under conditions where KGs' semantics and structure are deliberately perturbed and contaminated. In both settings, we measure the degree of hallucination and the overall quality of the responses (see §5.1 for details).

Based on our experiments, we find that (1) integrating KG information generally mitigates factual errors, especially for queries requiring deep reasoning; (2) directly performing reasoning in the LLM itself (e.g., internal reasoning strategies like Chain-of-thought (Kim et al., 2023) and Self-Consistency (Wang et al., 2022)) may cause bias and hallucination; (3) subgraph-based methods often achieve the best performance for simpler query types; and (4) incorporating KGs effectively reduces hallucinations in LLMs even when the KG is partially contaminated.

## 2 RELATED WORK

Due to the stochastic decoding process of Large Language Models (LLMs), *i.e.*, sampling the next token in the sequence, LLMs exhibit probabilistic behaviors: (1) potentially yielding varied outputs of the same input across different instances (Agrawal et al., 2023); (2) cannot accurately interpret phrases or terms when the context is vague and resides in a knowledge gap of the model. This will lead to outputs that may sound plausible but are often irrelevant or incorrect. This will lead to outputs that may sound plausible but are often irrelevant or incorrect. This "hallucinations" undermines the reliability of LLMs (Huang et al., 2023). One emerging research trend is enhancing LLMs through integrating external knowledge graphs (Agrawal et al., 2023). KGs offer structured, explicit, and up-to-date factual knowledge, including domain-specific knowledge, providing a faithful knowledge source for reasoning (Zheng et al., 2023; Agrawal et al., 2023; Sui et al., 2022). Moreover, each piece of information in KGs can be traced back to its source, providing context and provenance. This traceability not only aids in verifying the reliability of the information but also provides clear pathways of reasoning, making the interpretation process transparent.

Researchers employ diverse strategies to augment the LLMs by integrating external KGs (Sui et al., 2024; He et al., 2024b). For example, KAPING (Baek et al., 2023b) matches entities in questions to retrieve related triples from knowledge graphs for zero-shot question answering. Wu et al. (2023) finds that converting these triples into textualized statements can further enhance LLM performance. StructGPT (Jiang et al., 2023b) propose to convert user query into structured formats (e.g., SPARQL) for information extraction from KGs. Following the succuess of internal reasoning-enhancement methods like Chain-of-thoughts (CoT) (Wei et al., 2022), Reflexion (Shinn et al., 2024), and Tree-of-thoughts (ToT), He et al. (2022) propose "rethinking with retrieval" to use decomposed reasoning

steps from CoT prompting to retrieve external knowledge, leading to more accurate and faithful explanations. IR-CoT (Trivedi et al., 2022b) interleaves the generation of CoT with knowledge retrieval from corresponding KGs, iteratively guiding both retrieval and reasoning for multi-step questions. MindMap (Wen et al., 2023) introduce a plug-and-play approach to evoke graph-of-thoughts reasoning in LLMs. Similarly, RoG (Luo et al., 2023b) use KGs to create faithful reasoning paths based on various relations, enabling interpretable reasoning in LLMs.

However, current benchmarks for testing the capabilities of these LLM+KG models are predominantly closed-ended, restricting responses to a limited set of entities/relations or a set of logical forms derived from specific facts of KG. Hence, they can only test a very limited subset of the LLM's tendency to hallucinate, leaving a gap in the assessment of complex, real-world scenarios. Particularly, standard metrics such as FActScore (Min et al., 2023) and SAFE (Wei et al., 2024) for evaluating the hallucination rate of LLMs require open-ended settings, *i.e.*, questions are phrased as a statement which requires a longer answer. Compared with previous works, our proposed OKGQA is tailored for evaluating LLMs enhanced with KGs under open-ended, real-world question-answering scenarios. The benchmark extends the assessment of closed-ended question answering to an open-ended setting, which can further support the assessment of hallucination of LLMs.

## 3 OKGQA: AN OPEN-ENDED KNOWLEDGE GRAPH QUESTION-ANSWERING BENCHMARK

OKGQA is a comprehensive benchmark designed to assess how effectively LLMs enhanced with KGs perform in open-ended, real-world-like question answering scenarios. Unlike existing benchmarks that focus primarily on closed-ended tasks, OKGQA presents diverse open-ended question types that mirror the variable nature of practical applications. As illustrated in Figure 1, given a complex query and its corresponding subgraph in a KG, the system must be capable of understanding the relationships within the data and performing human-like reasoning over the KG content to compose a paragraph-long answer. **In the following section**, we first describe our dataset construction, including query generation via LLM templates and KG subgraph extraction with PPR pruning. We then introduce OKGQA-P, a benchmark variant designed to evaluate model robustness under KG perturbations, detailing our perturbation methods and the metrics used to assess semantic and structural deviations. Due to the page limitation, we also include some extension of our benchmark in Appendix §C, including multilingual setup, and more analysis.

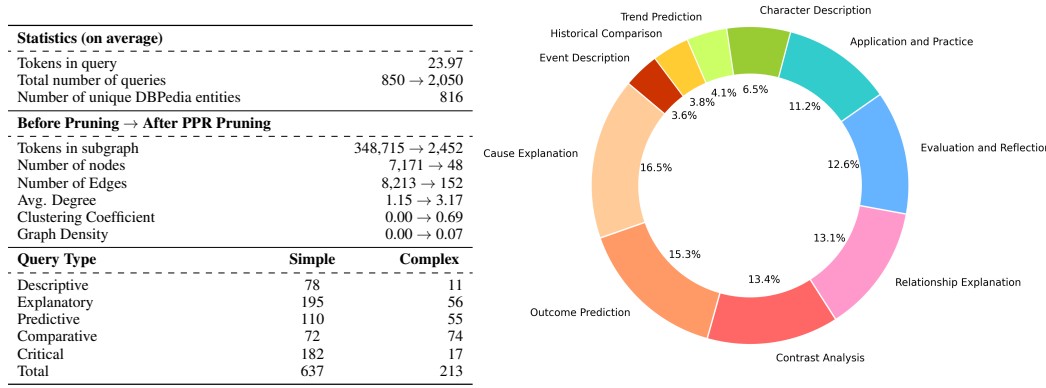

|  (a) Dataset statistics and query types | (b) Distribution of sub-query types |

Figure 2: (left) Dataset statistics and query types, (right) Sub-query type distribution

### 3.1 DATASET CONSTRUCTION

**Queries.** We utilize a template-based approach to generate a diverse range of queries using LLMs, including categories such as descriptive, explanatory, predictive, comparative, and critical queries. Details regarding specific templates and example queries can be found in Table 1, while the corresponding prompts are provided in the Appendix E. To ensure that the generated queries represent

real-world scenarios and complexities, we employ an iterative optimization approach that utilizes both automated and human evaluation to refine the query generation process (the details are given in Appendix B.1). Initially, we generate a diverse set of query candidates from a seed instruction. These candidates undergo automated evaluation using an LLM-based evaluator, which assigns quality scores $s_{\text{auto}}$ on a scale of 1-10, with higher scores indicating better performance across multiple metrics. Subsequently, human evaluators assess the same queries, producing corresponding normalized scores $s_{\text{human}}$ within the same range. To refine the dataset, we iteratively optimize the input instructions by minimizing the discrepancy between $s_{\text{human}}$ and $s_{\text{auto}}$. This optimization process ensures alignment between automated and human quality assessments. The queries are also categorized by complexity with detailed statistics in Figure 2.

| Type | Sub-Type | Description / Template | Example |
|------|----------|------------------------|---------|
| Descriptive | Character Description | Describe a [person]'s significant contributions during their career. | Please describe Albert Einstein's contributions to the field of physics. |
| | Event Description | Provide a detailed description of the background and course of an [event]. | Please provide a detailed description of the background and course of the French Revolution. |
| Explanatory | Cause Explanation | Why did [person] take [action] at [time]? | Why did Nixon choose to resign from the presidency in 1974? |
| | Relationship Explanation | Explain the relationship between [entity A] and [entity B] and its significance. | Explain the relationship between Alexander the Great and Aristotle and its significance. |
| Predictive | Trend Prediction | Based on the historical behavior of [entity], what do you think it might do in the future? | Based on Tesla's historical behavior, in which fields do you think it might innovate in the future? |
| | Outcome Prediction | Based on the current situation, how do you predict [event] will develop? | Based on the current international situation, how do you predict climate change policies will develop? |
| Comparative | Contrast Analysis | Compare and contrast the similarities and differences between [entity A] and [entity B] in [aspect]. | Compare and contrast the leadership styles of Steve Jobs and Bill Gates. |
| | Historical Comparison | Compare the impact of [historical event A] and [historical event B]. | Compare the impact of World War I and World War II on the global order. |
| Critical | Evaluation and Reflection | How do you evaluate the impact of [person/event] on [field]? Please explain your viewpoint. | How do you evaluate Martin Luther King's impact on the civil rights movement? Please explain your viewpoint. |
| | Application and Practice | How do you think [theory/method] can be applied to [practical issue]? | How do you think machine learning technology can be applied to medical diagnostics? |

Table 1: Query types and examples in OKGQA. **Brown** is used to highlight the placeholders (*e.g.*, [person], [event]) in description, while **Teal** highlights the specific entities to replace the placeholders.

**KG Sub-graphs.** To reduce the size of KGs while retaining relevant information, we follow previous work (Yih et al., 2016; Talmor & Berant, 2018) by sampling subgraphs from DBpedia (Noted that all queries in OKGQA can be answered using DBpedia). We extract all triples contained within the $K$-hop neighbors from the entities mentioned in the query. We set $K = 2$ to balance graph size and computational feasibility. As increasing beyond 2-hop subgraphs generally leads to exponential growth in edges and nodes (Jin et al., 2020), which increase excessive noise and complicating information retrieval[2]. To further reduce the size of the 2-hop subgraphs, we leverage Personalized Page-Rank (PPR) (Bahmani et al., 2010) to prune the nodes/edges that are not relevant to the query (the details of the PPR algorithm are discussed in Appendix B.2). We compare the statistics of subgraphs before and after PPR pruning in Figure 2a.

## 3.2 OKGQA-P: Benchmark with Noise & Perturbations in KGs

KGs are often annotated by humans and can contain errors such as mislabeled attributes or missing relations. To mimic the real situations where KGs' quality may not be fully reliable, we propose **OKGQA-P** to assess the model performance under deliberately perturbed and contaminated KGs. We introduce various perturbation scenarios including mislabeled attributes, incorrect relations, and missing connections to test how well models can handle flawed or incomplete KG data. To quantify the degree of perturbation, we measure the semantic and structural similarity between the original and the modified KG as defined below.

**Notation.** Let $\mathcal{F}_\theta$ be a KG-augmented model, and KG as $\mathcal{G} = (\mathcal{V}, \mathcal{E}, \mathcal{T})$, where $\mathcal{V}$ is the set of entities (nodes), $\mathcal{E}$ is the set of relation types (edges), and $\mathcal{T} = \{(v_1, e, v_2)|v_1, v_2 \in \mathcal{V}, e \in \mathcal{E}\}$ is the set of triplets composed of entities and relations. Let $\mathcal{G}' = (\mathcal{V}, \mathcal{E}', \mathcal{T}')$ be the KG after perturbing $\mathcal{G}$, where $\mathcal{E}' \neq \mathcal{E}$ and $\mathcal{T}' \neq \mathcal{T}$. Let $f(\mathcal{G}, \mathcal{G}')$ be a function that measures the similarity between $\mathcal{G}$ and $\mathcal{G}'$. Let $g(\mathcal{G})$ be the downstream performance when evaluating $\mathcal{F}_\theta$ on data samples $X$ and $\mathcal{G}$.

---

[2]This choice is also informed by common practices in other benchmarks, such as WebQSP (Yih et al., 2016) and CWQ (Talmor & Berant, 2018), where 2-hop subgraphs are widely used for similar KGQA tasks.

**High-level Procedure.** First, we test $\mathcal{F}_\theta$ on data samples $X$ and $\mathcal{G}$ to get the original performance $g(\mathcal{G})$. Second, we perturb $\mathcal{G}$ to obtain $\mathcal{G}'$. Third, we evaluate $\mathcal{F}_\theta$ on data samples $X$ and $\mathcal{G}'$ to get the perturbed performance $g(\mathcal{G}')$. Finally, we measure $g(\mathcal{G}) - g(\mathcal{G}')$ and $f(\mathcal{G}, \mathcal{G}')$ to assess how robust $\mathcal{F}_\theta$ is, *i.e.*, to assess the model performance under conditions where KGs' semantics and structure are deliberately perturbed.

To quantify how much the perturbed KG has deviated from the original KG, *i.e.*, $f(\mathcal{G}, \mathcal{G}')$, we leverage metrics from (Raman et al., 2020) for capturing semantics (ATS) and structural (SC2D, SD2) similarity between KGs. Intuitively, ATS leverages a pre-trained LM for link prediction to measure the probability of each edge from $\mathcal{G}'$ existing in $\mathcal{G}$, while SC2D and SD2 measure the structural similarity between two KGs based on local clustering coefficient and degree distribution. For each of the three metrics, higher value indicates higher similarity. The detailed description can be found in Appendix B.5, with visualization in Figure 5.

For the perturbation methods, we consider four edge-based perturbation heuristics based on (Raman et al., 2020) as follows:

- **Relation Swapping (RS)** randomly chooses two edges from $\mathcal{T}$ and swaps their relations.
- **Relation Replacement (RR)** randomly chooses an edge $(v_1, e, v_2) \in \mathcal{T}$, and replaces the $e_1$ with another relation $e_2 = \operatorname{argmin}_{e \in \mathcal{E}} S_\mathcal{G}(v_1, e, v_2)$, where $S_\mathcal{G}(\cdot)$ is a KG score function adapted from ATS. This yield "harder negatives" - triplets that are semantically similar but incorrect.
- **Edge Rewiring (ER)** randomly chooses an edge $(v_1, e, v_2) \in \mathcal{T}$, then replaces $v_2$ with another entity $v_3 \in \mathcal{E} \backslash \mathcal{N}_1(v_1)$, where $\mathcal{N}_1(v_1)$ represents the 1-hop neighborhood of $v_1$.
- **Edge Deletion (ED)** randomly chooses an edge $(v_1, e, v_2) \in \mathcal{T}$ and deletes it.

We control perturbation level by adjusting the percentage of edges in $\mathcal{G}$ that are perturbed. Refer to Figures 5 and 6 for empirical results.

# 4 EXPLORING KG-AUGMENTED FRAMEWORK FOR REDUCING HALLUCINATION

To explore whether KG-augmented approaches can mitigate LLMs' hallucination, we propose a unified framework as shown in Figure 3. Our framework follows the paradigm of retrieval augmented generation (RAG) (Edge et al., 2024; Baek et al., 2023a), which retrieves essential information from the KGs, and then uses the retrieved knowledge to enhance the LLM's generation (§4.1). It consists of two components, *i.e.*, `Graph-guided retrieval` (§4.2) and `Graph-guided generator` (§4.3), with a variety of algorith-

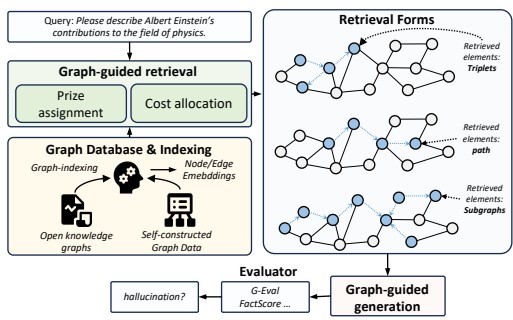

Figure 3: Overview of KG-augmented framework.

mic design choices. We analyze the strategies within each component in §5, aiming to shed light on the best practices for leveraging KGs for reducing hallucinations in LLMs.

## 4.1 FORMALIZATION

We formalize the KG-augmented framework for reducing hallucination as follows. Given a user query $q$, a pretrained language model generates a paragraph-like answer $a$ by modeling the conditional probability $p(a|q)$. To explore whether KGs help reduce hallucinations of LLMs, we introduce retrieved knowledge $\mathcal{Z}$ from the KG and define:

$$p(a|q) = \sum_{\mathcal{Z} \subseteq \mathcal{G}} p_\phi(a|q, \mathcal{Z}) p_\theta(\mathcal{Z}|q, \mathcal{G}), \tag{1}$$

where $p_\phi(a|q, \mathcal{Z})$ is the likelihood of generating the paragraph-like answer $a$ conditioned on $q$ and $\mathcal{Z}$ (parameterized by $\phi$), and $p_\theta(\mathcal{Z}|q, \mathcal{G})$ models the retrieval of knowledge subsets (parameterized by $\theta$). Because the number of possible subsets $\mathcal{Z}$ can be exponentially large relative to the size of $\mathcal{G}$, we

approximate the sum by selecting the most probable knowledge subset: $\mathcal{Z}^* = \text{argmax}_{\mathcal{Z}\in\mathcal{G}}p_\theta(\mathcal{Z}|q,\mathcal{G})$, yielding:

$$p(a|q) \approx p_\phi(a|q,\mathcal{Z}^*)p_\theta(\mathcal{Z}^*|q,\mathcal{G}) \tag{2}$$

## 4.2 GRAPH-GUIDED RETRIEVAL (G-RETRIEVAL)

Our goal in G-retrieval is to extract a compact yet informative subset $\mathcal{Z}^*$ from the KG that best supports answering the user query $q$. We first encode the query and all KG elements (nodes/edges) into a unified embedding space using a language model. We then measure the relevance of each element to $q$ (e.g., via cosine similarity) and identify a set of top-$k$ nodes and edges for the query.

To balance retrieving as many relevant nodes and edges as possible while keeping the $\mathcal{Z}^*$ size manageable, we adopt a **prize-cost trade-off strategy** (Balas, 1989) to guide the retrieval process: (1) *Prize assignment*: based on the computed similarity scores, we assign prizes to nodes and edges to quantify their relevance to the query. Specifically, we assign the top-$k$ nodes/edges with descending prize values from $k$ to 1, while nodes and edges outside the top-$k$ receive a prize of 0. Formally: $p_v = \max(0, k - \text{rank}(v) + 1)$ and $p_e = \max(0, k - \text{rank}(e) + 1)$. (2) *Cost allocation*: to manage the retrieved knowledge size, we assign penalties as cost $C_e$ during the expansion of the retrieved paths or subgraphs. The final retrieval process aims to maximize the total prize (i.e., relevance) while minimizing associated costs.

We explore three retrieval variants for G-retrieval design (e.g., triplets, paths and subgraphs) as demonstrated in Figure 3.

- **Triplet-retrieval**: retrieves a fixed number of triplets with the highest total prize assigned to their respective triplets.
- **Path-retrieval**: starting from a fixed number of $k$ of high-prize nodes, we greedily expand paths $\mathcal{P} = \{v_1, e_1, v_2, \ldots, e_{n-1}, v_n\}$ to maximize score: $S(\mathcal{P}) = \sum_{i=1}^{n} p_{v_i} + \sum_{i=1}^{n-1} p_{e_i} - \sum_{i=1}^{n-1} c_e$. We use a priority queue to iteratively return paths with top-scores and subject to maximum lengths and cycles. The details of path-retrieval can be found in Appendix B.3.
- **Sub-graph retrieval**: building on He et al. (2024a), we use the Prize-Collecting Steiner Tree (PCST) algorithm to find a connected subgraph $\mathcal{S}$ that maximizes $S(\mathcal{S}) = \sum_{n\in V_\mathcal{S}} p_{v_i} + \sum_{e\in E_\mathcal{S}} p_{e_i} - \sum_{e in E_\mathcal{S}} c_e$. Unlike in path-retrieval, we only yield one subgraph that maximizes the total score.

## 4.3 GRAPH-GUIDED GENERATION (G-GENERATOR)

After retrieving $\mathcal{Z}^*$, the G-Generator use this knowledge to generate the paragraph-like response the user query. The generation is modeled as a sequential decision-making process: at each time step $t$, token $a_t$ is generated conditioned on $q$, $\mathcal{Z}^*$, and the previously generated tokens $a_{0:t-1}$:

$$p(a|q,\mathcal{Z}^*) = \prod_{t=1}^{T} p_\theta(a_t|q,\mathcal{Z}^*,a_{0:t-1}), \tag{3}$$

where $\theta$ denotes the parameters of a neural text generation model. The generation stops when an end-of-sequence token is produced or when the maximum sequence length $T$ is reached.

## 5 EXPERIMENTS

In this section, we first introduce the evaluation metrics, and then focus on two main research questions: RQ1: Can KGs reduce hallucination in LLMs? and RQ2: How are KG-Aware methods affected by noise/perturbations in KGs?

## 5.1 EVALUATION METRICS & SETUP

We quantify LLM hallucinations using two public metrics: **FActScore** (Min et al., 2023) and **SAFE** (Wei et al., 2024). **FActScore** measures factual precision by decomposing a long-form text into atomic facts and validating each against a reliable knowledge base like Wikipedia. In contrast, **SAFE** employs a language model as an investigative agent that iteratively employs Google Search

queries to assess whether search results support the fact. For both metrics, we report the proportion of supported atomic facts out of the total atomic facts extracted from LLM responses.

In addition to the hallucination metrics, we propose four metrics using LLM-as-evaluator (Li et al., 2024) to quantify the quality of generated responses from LLM (Edge et al., 2024; Wang et al., 2023). In specific, we use G-Eval (Liu et al., 2023) framework for the evaluation and provide relevant Wikipedia pages of each query as context to enhance G-Eval's robustness and stability. The four metrics are defined as follows: **Context Relevance**: measures how well the generated response aligns with the provided context. **Comprehensiveness**: assesses how thoroughly the answer addresses all aspects and details of the question. **Correctness**: measures the clarity and specificity of the generated answer to the question. **Empowerment**: evaluates how well the generated answer helps the reader understand the topic and make informed decisions. The detailed prompt can be found in Appendix E.

We use gpt-4o-mini (from November 2024 to January 2025) as LLM backbone for all the evaluation metrics. As using LLM-as-evaluator frameworks may raise concern regarding **potential self-enhancement** or bias from the selection of the models (Gu et al., 2024; Li et al., 2024), we conduct additional validation in Appendix B.4 (including human evaluation alignment and cross-validation across different LLM backbones), and find that the choice of LLM in the LLM-as-evaluator framework has little impact on the overall evaluation and demonstrate high correlation with the human evaluation, supporting the reliability of our testing.

For other testing LLM backbones mentioned in this section, we consider a range of widely used LLMs of different scales, including GPT-4o, GPT-4o-mini (from November 2024 to January 2025), Llama-3.1-8B-instruct (Dubey et al., 2024), Mistral-7B-instruct-v0.3 (Jiang et al., 2023a), and Gemma-2-9B-it (Team et al., 2024). Considering the trade-off between cost and performance, we use text-embedding-3-small model from OpenAI (from November 2024 to January 2025) as embedding model for G-retrieval process.

## 5.2 RQ1: MAIN RESULTS - CAN KGS REDUCE HALLUCINATION IN LLMS?

To explore whether KGs can help reduce hallucination in LLMs, we benchmark the LLMs in different settings. We use zero-shot and few-shot prompting as baselines without injecting external knowledge. In addition, we consider leveraging LLMs' internal knowledge to do Chain-of-thought (Kim et al., 2023), or self-consistency (Wang et al., 2022), and more general RAG systems like IRCoT (Trivedi et al., 2022a) which retrieves paragraphs from Wikipedia to augment CoT generation. For LLMs augmented with KGs, we consider three KG retrieval variants: triplets, paths, and subgraphs to study the impact of G-retrieval for reducing LLMs' hallucinations. The results are shown in Table 2 and Figure 4. We obtain some intriguing findings:

**Retrieving KG information can indeed mitigate factual errors in the responses.** Methods integrating knowledge extracted from KGs show clear improvements in factual accuracy and comprehension scores compared to the baselines. For example, under Var-2 (triplet retrieval), GPT-4o achieves a FActScore of 72.55% ± 0.85%, which is a significant increase over the baseline score of 55.35% ±0.95%. Moreover, these methods can be combined with strategies like CoT+SC, enhancing response quality with minimal increase in hallucination ratio. The radar chart in Figure 4 further emphasizes that in most query types, integrating knowledge retrieved from KGs mitigates the hallucination issue compared to baselines, particularly in query types such as "Evaluation and Reflection," "Outcome

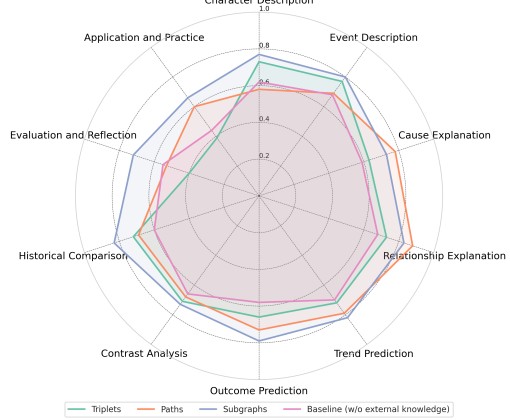

Figure 4: Comparison results of different forms of information over different queries.

Prediction," and "Cause Explanation," which require more reasoning and analysis rather than merely listing information. The findings also apply to open-source models like mistral-7B-Instruct-v0.3 and Llama-3.1-8B-instruct, illustrating the consistency of the finding. In addition, compared with RAG method IRCoT (Trivedi et al., 2022b), leveraging Wikipedia documents, improves performance over

zero-shot and 4-shot prompting by providing broad contextual support, it struggles with correctness and hallucination control due to the potential introduction of irrelevant or conflicting information. Our KG-based methods consistently outperform IRCoT, particularly in correctness, SAFE, and FActScore.

| Models | G-Eval | | | | Hallucination | |
|---|---|---|---|---|---|---|
| | Context Relevance | Comprehensiveness | Correctness | Empowerment | SAFE | FActScore |
| **Baseline: Without External Knowledge (Zero-shot prompting)** | | | | | | |
| GPT-4o | 68.12% ± 0.88% | 65.41% ± 0.79% | 60.41% ± 0.38% | 62.41% ± 0.84% | 82.47% ± 0.62% | 55.34% ± 0.93% |
| GPT-4o-mini | 63.21% ± 0.49% | 60.11% ± 0.47% | 55.43% ± 0.63% | 58.72% ± 0.62% | 80.14% ± 0.89% | 50.23% ± 1.01% |
| llama-3.1-8b-instruct | 57.12% ± 0.91% | 54.74% ± 1.20% | 49.01% ± 0.61% | 52.21% ± 0.71% | 79.33% ± 0.91% | 45.14% ± 0.32% |
| mistral-7B-Instruct-v0.3 | 55.71% ± 1.21% | 52.00% ± 1.31% | 47.03% ± 0.94% | 50.13% ± 1.04% | 78.27% ± 0.83% | 44.37% ± 1.23% |
| gemma-2-9b-it | 53.63% ± 1.33% | 50.00% ± 1.33% | 45.72% ± 0.71% | 48.15% ± 0.93% | 77.11% ± 0.78% | 40.94% ± 0.83% |
| **Baseline: Without External Knowledge (4-shot prompting)** | | | | | | |
| GPT-4o | 70.61% ± 0.62% | 67.43% ± 0.81% | 62.33% ± 0.37% | 64.51% ± 0.12% | 83.39% ± 0.53% | 57.45% ± 0.78% |
| GPT-4o-mini | 65.53% ± 0.94% | 62.33% ± 1.03% | 57.23% ± 0.68% | 60.47% ± 0.83% | 81.62% ± 0.69% | 52.34% ± 0.76% |
| llama-3.1-8b-instruct | 59.43% ± 0.32% | 56.31% ± 0.78% | 51.27% ± 0.32% | 54.33% ± 0.41% | 80.27% ± 0.78% | 47.24% ± 1.03% |
| mistral-7B-Instruct-v0.3 | 57.34% ± 1.04% | 54.13% ± 1.31% | 49.27% ± 0.84% | 52.46% ± 0.94% | 79.12% ± 0.87% | 45.13% ± 1.42% |
| gemma-2-9b-it | 55.24% ± 1.49% | 52.27% ± 1.21% | 47.14% ± 0.36% | 50.36% ± 0.51% | 78.00% ± 0.77% | 44.32% ± 1.58% |
| **Baseline: With Wikipedia documents** | | | | | | |
| GPT-4o - IRCoT | 73.12% ± 0.32% | 69.23% ± 0.42% | 66.33% ± 0.34% | 65.51% ± 0.11% | 87.39% ± 0.68% | 69.45% ± 0.34% |
| GPT-4o-mini - IRCoT | 70.31% ± 0.32% | 64.42% ± 1.31% | 61.37% ± 0.48% | 63.89% ± 0.72% | 84.72% ± 0.48% | 65.72% ± 1.03% |
| **Var-1: With CoT Prompting** | | | | | | |
| GPT-4o - CoT | 72.76% ± 0.92% | 69.56% ± 0.74% | 64.48% ± 0.63% | 66.69% ± 0.69% | 80.07% ± 0.83% | 54.30% ± 0.87% |
| GPT-4o - CoT+SC | 75.81% ± 0.65% | 71.62% ± 0.74% | 66.55% ± 0.75% | 68.74% ± 0.15% | 79.03% ± 0.48% | 53.23% ± 0.78% |
| llama-3.1-8b-instruct - CoT+SC | 63.69% ± 0.32% | 60.44% ± 0.59% | 55.46% ± 0.52% | 58.53% ± 1.11% | 76.00% ± 0.63% | 45.05% ± 0.97% |
| mistral-7B-Instruct-v0.3 - CoT+SC | 61.35% ± 0.93% | 58.33% ± 1.02% | 53.42% ± 0.79% | 56.47% ± 0.85% | 74.30% ± 0.21% | 42.00% ± 0.29% |
| gemma-2-9b-it - CoT+SC | 59.42% ± 0.27% | 56.27% ± 0.84% | 51.34% ± 1.42% | 54.34% ± 1.31% | 71.09% ± 0.43% | 39.85% ± 1.03% |
| **Var-2: With Triplets Extracted from KGs Provided** | | | | | | |
| GPT-4o | 74.62% ± 0.65% | 70.44% ± 0.79% | 65.37% ± 0.72% | 67.12% ± 0.71% | 89.20% ± 1.42% | 72.53% ± 0.83% |
| GPT-4o-mini | 69.50% ± 0.81% | 65.03% ± 0.92% | 60.21% ± 0.65% | 63.43% ± 1.01% | 87.52% ± 0.34% | 67.73% ± 0.95% |
| llama-3.1-8b-instruct | 63.45% ± 1.13% | 59.33% ± 1.05% | 54.23% ± 0.75% | 57.33% ± 0.12% | 85.37% ± 0.72% | 62.37% ± 0.82% |
| mistral-7B-Instruct-v0.3 | 61.34% ± 0.31% | 57.21% ± 0.89% | 52.29% ± 0.32% | 55.12% ± 0.43% | 84.21% ± 0.84% | 60.28% ± 1.05% |
| gemma-2-9b-it | 59.25% ± 1.06% | 55.29% ± 0.44% | 50.15% ± 0.85% | 53.73% ± 0.95% | 83.18% ± 0.43% | 58.13% ± 0.91% |
| GPT-4o - CoT+SC | 76.71% ± 0.53% | 72.34% ± 0.21% | 67.33% ± 1.31% | 69.64% ± 0.33% | 88.11% ± 0.57% | 71.45% ± 0.53% |
| **Var-3: With Paths Extracted from KGs Provided** | | | | | | |
| GPT-4o | 78.71% ± 0.53% | 74.53% ± 0.31% | 69.42% ± 0.23% | 71.63% ± 0.61% | 90.20% ± 0.59% | **75.61% ± 0.51%** |
| GPT-4o-mini | 73.64% ± 0.93% | 69.41% ± 0.22% | 64.35% ± 0.72% | 67.52% ± 0.82% | 88.22% ± 0.34% | 70.53% ± 0.24% |
| llama-3.1-8b-instruct | 67.51% ± 0.46% | 63.62% ± 1.39% | 58.41% ± 0.93% | 61.57% ± 0.94% | 86.33% ± 0.94% | 65.42% ± 0.95% |
| mistral-7B-Instruct-v0.3 | 65.48% ± 0.94% | 61.37% ± 1.01% | 56.34% ± 0.23% | 59.45% ± 0.43% | 85.26% ± 0.85% | 63.31% ± 1.33% |
| gemma-2-9b-it | 63.35% ± 1.37% | 59.23% ± 0.91% | 54.31% ± 0.91% | 57.41% ± 0.27% | 84.13% ± 0.21% | 61.23% ± 1.04% |
| GPT-4o - CoT+SC | 80.87% ± 0.42% | 76.60% ± 0.65% | 71.54% ± 0.53% | 73.79% ± 1.21% | 89.11% ± 0.63% | 74.53% ± 0.24% |
| **Var-4: With Subgraphs Extracted from KGs Provided** | | | | | | |
| GPT-4o | 80.81% ± 0.43% | 76.63% ± 0.65% | 71.57% ± 0.51% | 73.70% ± 0.62% | **90.83% ± 0.63%** | 75.33% ± 0.29% |
| GPT-4o-mini | 75.70% ± 0.44% | 71.51% ± 0.83% | 66.43% ± 0.76% | 69.60% ± 0.65% | 88.71% ± 0.72% | 70.12% ± 0.87% |
| llama-3.1-8b-instruct | 69.61% ± 0.84% | 65.45% ± 0.93% | 60.41% ± 0.65% | 63.42% ± 0.15% | 86.12% ± 0.35% | 65.44% ± 0.87% |
| mistral-7B-Instruct-v0.3 | 67.55% ± 0.87% | 63.35% ± 0.43% | 58.37% ± 0.71% | 61.45% ± 0.32% | 85.21% ± 0.81% | 63.12% ± 0.94% |
| gemma-2-9b-it | 65.45% ± 0.95% | 61.23% ± 1.0% | 56.31% ± 0.35% | 59.40% ± 0.85% | 84.51% ± 0.99% | 63.74% ± 0.49% |
| GPT-4o - CoT+SC | **82.90% ± 0.57%** | **78.72% ± 0.61%** | **73.64% ± 0.43%** | **75.80% ± 0.75%** | 89.12% ± 0.94% | 75.42% ± 1.31% |

Table 2: Comparison results of various forms of information extracted from the KGs.

**Directly performing reasoning in the LLM itself does not mitigate hallucinations.** We benchmark the hallucination ratio of LLMs using internal reasoning strategies like CoT and Self-consistency, as shown in Var-1 in Table 2. It shows that these methods can improve response quality (i.e., G-Eval) compared to baselines, but do not consistently improve factuality, and sometimes even diminish. This shows that relying solely on internal reasoning is inadequate for mitigating hallucinations, highlighting the necessity for external knowledge to address this issue effectively.

**Subgraph retrieval generally achieves best performance across different query types, especially for simpler queries.** We demonstrate the performance of different retrieval methods across different query types in Figure 4, showing that subgraphs achieve the best performance. Especially for simpler queries ("Character Description" and "Event Description" which do not require intensive reasoning). Even for queries like "Relationship Explanation" and "Cause Explanation" which require stepwise reasoning, subgraph methods still demonstrate promising performance. This suggests that while different forms of retrieved knowledge offer unique benefits for specific types of queries, subgraphs provide consistently good performance.

## 5.3 RQ2: How Are KG-Aware Methods Affected by Noise / Perturbations in KGs?

We benchmark different KG-augmented LLMs on our OKGQA-P setting, where we deliberately perturb and contaminate the semantics and structure of KGs to simulate the real-world situation where KGs may not have high quality. Specifically, we consider different perturbation methods discussed in §3.2 and control the perturbation level based on the percentage of KG edges being perturbed. We first illustrate how much the perturbed KG has been deviated from the original KG with the increase of perturbation level, shown in Figure 5. It shows that the perturbation methods like edge deletion, rewiring and swapping have relatively weak influence on ATS (which intuitively measures

semantic similarity), even as the perturbation level increases. For the edge deletion methods, only if the perturbation level reaches 1.0, the ATS goes to 0, otherwise, the ATS remains higher compared to other settings.

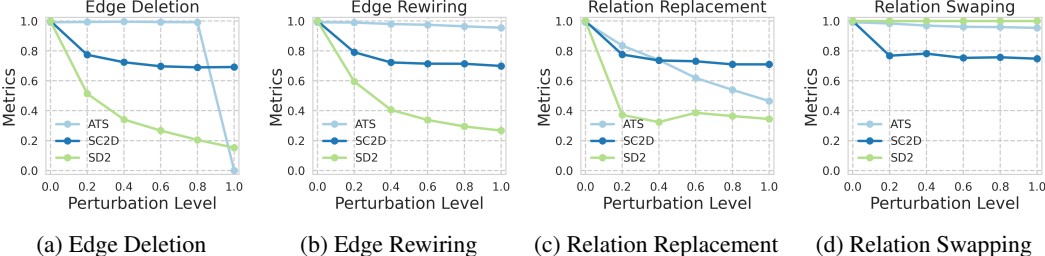

| (a) Edge Deletion | (b) Edge Rewiring | (c) Relation Replacement | (d) Relation Swaping |

Figure 5: Performance Metrics (ATS, SC2D, SD2) vs. Perturbation Level for Different Perturbation Methods.

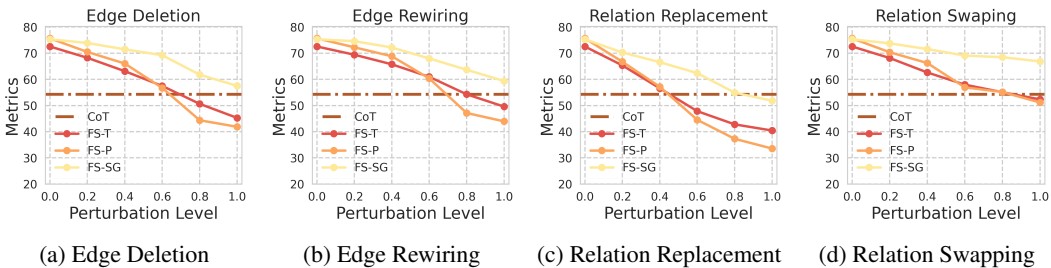

| (a) Edge Deletion | (b) Edge Rewiring | (c) Relation Replacement | (d) Relation Swaping |

Figure 6: Performance Metric (FActScore) vs. Perturbation Level for Different Perturbation Methods and Different Retrieval Methods. **FS-T** refers to FActScore metric using triplets, **FS-P** refers to using paths, and **FS-SG** refers to using sub-graphs.

Figure 6 illustrates the hallucination ratio using different methods on the perturbed KGs. We observe that (1) FS-SG consistently outperforms FS-T and FS-P even at higher perturbation levels, demonstrating its robustness by maintaining higher scores as perturbations increase. (2) FS-T and FS-P exhibit similar trends, each showing a significant performance drop as perturbation levels increase. Particularly, performance of FS-T and FS-P deteriorate when the perturbation level reaches 50%, *i.e.*, becoming worse than the baseline using CoT. (3) On the setting using Relation Replacement which severely harms the semantics of the KGs, FS-T and FS-P decline more sharply than FS-SG. However, it still outperforms the baseline when the perturbation level is smaller than 40%. **In summary**, we find that effectiveness of KG-derived information diminishes with a perturbation level at 50%, surpassing this level leads to a further decrease in performance. We think that before this perturbation level at 50%, incorporating external knowledge from KGs can mitigate hallucinations in LLMs compared to baseline using CoT. Considering practical scenarios, platforms like Wikidata are less likely to have perturbations as severe as 50% due to their ongoing updates and community-based quality control. This ensures the relevance and applicability of our findings in real-world settings.

## 6 CONCLUSION

In this paper, we propose OKGQA and variant OKGQA-P, to assess LLMs enhanced with KGs under open-ended, real-world question answering scenarios. Unlike existing benchmarks that focus primarily on closed ended tasks, OKGQA presents diverse open-ended question types that mirror the unpredictable nature of practical applications. We conduct a series of experiments and analyze the effectiveness of various retrieval methods and LLMs of different magnitudes, providing insights for further research. Our results underscore the significance of integrating KGs with LLMs to help reduce hallucination of LLMs, even in circumstances where the KGs are contaminated.

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

## A    APPENDIX

## B    IMPLEMENTATION DETAILS

### B.1    QUERY CONSTRUCTION

In this section, we discuss the details of the query construction of OKGQA. We first introduce the human-in-the-loop process to optimize the instruction for generating the queries, as shown in §B.1.1. We then present the metrics for quantify the generated queries in §B.1.2. Subsequently, we provide experiments results of human-in-the-loop process and demonstrate the Pearson correlation coefficients between human and LLM scores across rounds of optimization, and verify the inter-rater reliability across our evaluators in §B.1.3.

#### B.1.1    HUMAN-IN-THE-LOOP FOR INSTRUCTION OPTIMIZATION

To ensure that the generated queries accurately represent real-world scenarios and complexities, we propose a human-in-the-loop process to optimize the instruction used for generation, as shown in Figure 7. To ensure clarity, we summarize this optimization process here:

- Step 1: Generate a set of queries from an initial instruction.
- Step 2: Collect automatic evaluation scores $s_{\mathbf{auto}}$ by LLMs and human-label scores $s_{\mathbf{human}}$ by human annotators for these queries (normalized to the same range).
- Step 3: Identify patterns of discrepancies between these scores.
- Step 4: Let the LLM analysis the identified patterns to generate new instructions,

The step 3 and 4 are conducted by prompting LLM with prompt specified in §E.3, and steps 1 to 4 are running iteratively to reducing $s_{\mathbf{auto}}$ and $s_{\mathbf{human}}$ discrepancies. This process quite mimics the way of reinforcement learning with human feedback (RLHF) (Ouyang et al., 2022) and inherits the benefit that labeling the reward of the LLMs' output is much easier than labeling the output directly.

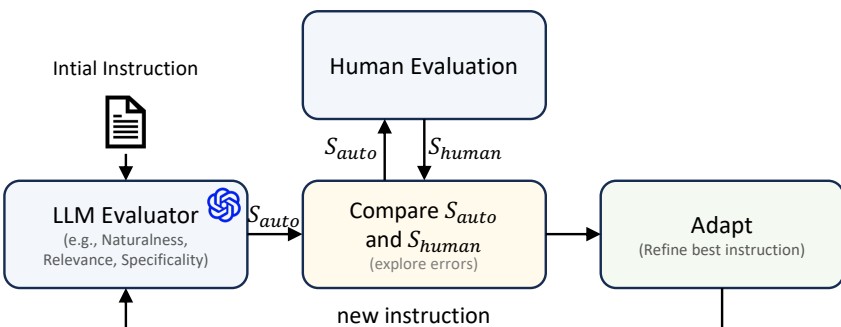

Figure 7: Human-in-the-loop of query construction.

#### B.1.2    METRICS FOR GENERATED QUERIES

We consider five metrics to measure the quality of the generated queries: (1) **Naturalness**: assessing how fluid and human-like the query sounds; (2) **Relevance**: measuring whether the query pertains directly to the entity and the context provided; (3) **Specificity**: determining the level of detail and granularity included in the query, ensuring it is not too broad or vague; (4) **Novelty**: evaluating the uniqueness of the query, ensuring it is not just a repetitive or common question; (5) **Actionability**: gauging whether the query prompts clear, definite answers or actions that are feasible within the given context. Each of these angles contributes to a holistic evaluation of the query's effectiveness and relevance in real-world applications.

### B.1.3 Verifying human-in-the-loop

For the human-label scores $s_{\mathbf{human}}$ collection, we have three evaluators participating in the manual assessment of query quality. All of the evaluators are computer science majors with fluent English skills. As the evaluation centers on various linguistic metrics such as naturalness, relevance, specificity, novelty, and actionability, we only require the evaluators to possess a fundamental understanding of English without restricting their majors. We calculate the Pearson correlation coefficients between human and LLM scores as shown in Table 3. It shows that as the rounds progress, agreement between humans and LLMs increases, suggesting that iterative feedback improves alignment between human annotation and LLM responses.

| Metric | Round 1 | Round 2 | Round 3 | Round 4 |
|---|---|---|---|---|
| Naturalness | 0.60 | 0.65 | 0.69 | 0.74 |
| Relevance | 0.55 | 0.59 | 0.64 | 0.70 |
| Specificity | 0.46 | 0.54 | 0.60 | 0.65 |
| Novelty | 0.49 | 0.57 | 0.63 | 0.67 |
| Actionability | 0.33 | 0.41 | 0.48 | 0.53 |

Table 3: Pearson correlation coefficients between human and LLM scores across rounds.

In addition, we also consider verifying the inter-rater reliability across three evaluators as shown in Table 4. We report the Cohen's Kappa coefficient for each pair of evaluators as follows. Using the (Landis & Koch, 1977) interpretation guidelines, the Cohen's Kappa coefficients for Naturalness and Relevance (ranging from 0.79 to 0.85) fall within the "Substantial" to "Almost Perfect" categories, indicating strong inter-rater reliability for these metrics. This reflects a shared understanding of the evaluation criteria, resulting in consistent ratings among evaluators. For Specificity, Novelty, and Actionability, the coefficients range from 0.58 to 0.68, placing them primarily in the "Moderate" to "Substantial" categories. These results suggest moderate reliability for these metrics, likely due to subjective interpretation and less clearly defined evaluation guidelines. Novelty, with lower coefficients around 0.61 to 0.63, highlights variability in ratings, suggesting that evaluators may have differing perspectives on what qualifies as novel (but the inter-rater reliability is still considered "Substantial"). Meanwhile, Actionability performs slightly better, nearing the "Substantial" range, indicating moderately consistent criteria.

| Metric | Evaluator 1 & 2 | Evaluator 1 & 3 | Evaluator 2 & 3 |
|---|---|---|---|
| Naturalness | 0.85 | 0.83 | 0.84 |
| Relevance | 0.81 | 0.79 | 0.80 |
| Specificity | 0.65 | 0.63 | 0.66 |
| Novelty | 0.60 | 0.58 | 0.61 |
| Actionability | 0.67 | 0.65 | 0.68 |

Table 4: Cohen's Kappa coefficient for various metrics.

### B.2 Personalized PageRank (PPR)

In this section, we discuss the details of the PPR algorithm used in §3.1 to prune the graph from DBPedia and concentrate on nodes most pertinent to the central nodes of interest. The PPR is calculated using the iterative formula:

$$\mathbf{p} = \alpha \mathbf{A}^\top \mathbf{p} + (1 - \alpha)\mathbf{s}, \tag{4}$$

where $\mathbf{p} \in \mathbb{R}^n$ is the PPR vector representing the relevance scores of $n$ nodes in the graph. $\alpha$ is the damping factor controlling the probability of continuing the random walk versus restarting from the personalization vector. $\mathbf{A}^\top$ is the transpose of the column-normalized adjacency matrix $\mathbf{A}$ of the graph, representing transition probabilities between nodes. $\mathbf{s} \in \mathbb{R}^n$ is the personalization vector, where we assign a value of 1 to the central nodes and 0 to all other nodes to emphasize their importance. To ensure convergence and computational efficiency, we set a tolerance parameter tol $= 1 \times 10^{-6}$ and a maximum iteration limit max_iter $= 100$. After computing the PPR vector $\mathbf{p}$, we apply a threshold of $1 \times 10^{-5}$ to prune the graph. Nodes with PPR scores below this threshold are considered insignificant with respect to the central nodes and are thus removed. This process effectively filters out less relevant nodes, resulting in a pruned graph that highlights the most significant relationships and structures pertinent to our analysis.

| Setting | Context Relevance | Comprehensiveness | Correctness | Empowerment | SAFE | FActScore |
|---|---|---|---|---|---|---|
| OKGQA (subgraphs) | $75.70\% \pm 0.44\%$ | $71.51\% \pm 0.83\%$ | $66.43\% \pm 0.76\%$ | $69.60\% \pm 0.65\%$ | $88.71\% \pm 0.72\%$ | $70.12\% \pm 0.87\%$ |
| + Multi-lingual context | $75.14\% \pm 0.33\%$ | $72.32\% \pm 0.19\%$ | $66.72\% \pm 0.74\%$ | $70.32\% \pm 0.57\%$ | $90.32\% \pm 0.48\%$ | $72.83\% \pm 0.93\%$ |

Table 5: Comparison of GPT-4o-mini Performance Using Monolingual and Multilingual Subgraphs

### B.3 PRIZE-COST-BASED PATH RETRIEVAL

In this section, we detail the path-retrieval method used in §4.2. It is designed to construct and evaluate paths in a graph based on predefined prize assignments and cost allocations. The objective is to form sequences of nodes and edges, represented as $\mathcal{P} = \{v_1, e_1, v_2, \ldots, e_{n-1}, v_n\}$, that maximize the overall score and minimize the costs. To efficiently manage the exploration of potential paths, we utilize a **priority queue**, a data structure that allows paths to be organized based on their scores, ensuring that the highest-scoring paths are processed first. The method starts by picking a number of starting nodes with high prizes. We then expand these starting points by exploring neighboring nodes. For each neighbor, the method calculates a new score. This score is the sum of the neighbor's prize and the edge's prize minus the edge's cost. If this neighbor hasn't been visited before, which helps avoid looping, the algorithm adds this neighbor to the path. This new path is then added to the priority queue. This expansion keeps going until the path reaches a maximum length or can't be extended further. The algorithm keeps track of paths already explored to avoid repetition and ensure paths don't loop back on themselves. When no more paths can be added or the priority queue is empty, the algorithm sorts the paths by their scores from highest to lowest.

### B.4 LLM EVALUATION CLARITY

To address the concern regarding potential self-enhancement bias in LLM-as-evaluator frameworks, we have conducted extensive validation of our evaluation approach. In specific, we randomly sample 100 questions and evaluated them using three different LLMs (gpt-4o-mini, llama-3.1-8b-instruct, and gemma-2-9b-it). We measured inter-model agreement using Cohen's Kappa as shown in Table 6, which showed substantial to almost perfect consistency. This indicates that the evaluation results are consistent across different LLMs, even when the model generating the responses is not the same as the one evaluating them (e.g., using gpt-4o-mini for generation and llama-3.1-8b-instruct for evaluation). These findings confirm that the evaluation is robust and independent of the specific LLM used as the evaluator.

| Metric | LLM 1 & 2 | LLM 1 & 3 | LLM 2 & 3 |
|---|---|---|---|
| G-Eval | 0.84 | 0.81 | 0.82 |
| FactScore | 0.78 | 0.74 | 0.78 |
| SAFE | 0.74 | 0.70 | 0.72 |

Table 6: Cohen's Kappa coefficient for different LLM pair comparisons. For the G-Eval, we use the average score of four sub-metrics for better readability. LLM 1: gpt-4o-mini; LLM 2: llama-3.1-8b-instruct; LLM 3: gemma-2-9b-it)

In addition, we also collect human evaluations for these 100 samples. Three experts annotators rate each anonymized response on context relevance, comprehensiveness, correctness and empowerment using a 1-5 Likert scale. The average human ratings are computed and compared with automated scores using G-Eval. The Pearson's correlation yields a score of 0.78, indicating strong alignment between human judgment and LLM-based evaluation. Combined with the inter-model agreement shown in Table 6, these results confirm that our evaluation is robust, consistent, and largely independent of the specific LLM used as the evaluator.

### B.5 KG SIMILARITY METRICS

In this section, we introduce the metrics used in §3 to measure the deviation of the perturbed KGs from the original KG. These metrics are adapted from (Raman et al., 2020) as presented below. ATS is mainly used to measure the semantic similarity between two KGs, while SC2D and SD2 are used to measure the structural similarity.

**Aggregated Triple Score (ATS):** ATS measures semantic similarity between two KGs. Let $s_{\mathcal{G}}$ be an edge (triple) scoring function, such that $s_{\mathcal{G}}(e_1, r, e_2)$ measures how likely edge $(e_1, r, e_2)$ is to exist in $\mathcal{G}$. Also, assume $s_{\mathcal{G}}$ has been pre-trained on $\mathcal{G}$ for link prediction. Then, ATS is defined as $f_{\text{ATS}}(\mathcal{G}, \mathcal{G}') = \frac{1}{|\mathcal{T}'|} \sum_{(e_1, r, e_2) \in \mathcal{T}'} s_{\mathcal{G}}(e_1, r, e_2) \in [0, 1]$, which denotes the mean $s_{\mathcal{G}}$ score across all edges in $\mathcal{G}'$. Intuitively, if a high percentage of edges in $\mathcal{G}'$ are also likely to exist in $\mathcal{G}$ (i.e., high ATS), then we say that $\mathcal{G}'$ and $\mathcal{G}$ have high semantic similarity. $s_{\mathcal{G}}$ is task-specific, as KGs from different tasks may differ greatly in semantics. We use the $s_{\mathcal{G}}$ from (Li et al., 2016); while ATS captures semantic KG differences, it is not sensitive to KG connectivity structure. Note that $f_{\text{ATS}}(\mathcal{G}, \mathcal{G})$ may not equal 1, since $s_{\mathcal{G}}$ may not perfectly generalize to KGs beyond those it was trained on.

**Similarity in Clustering Coefficient Distribution (SC2D):** SC2D measures structural similarity between two KGs and is derived from the local clustering coefficient (Saramäki et al., 2007; Onnela et al., 2005; Fagiolo, 2007). For a given entity in $\mathcal{G}$ (treated here as undirected), the local clustering coefficient is the fraction of possible triangles through the entity that exist (i.e., how tightly the entity's neighbors cluster around it). For entity $e_i \in \mathcal{E}$, the local clustering coefficient is defined as $c_i = 2\text{Tri}(e_i)/(\deg(e_i)(\deg(e_i) - 1))$, where $\text{Tri}(e_i)$ is the number of triangles through $e_i$, and $\deg(e_i)$ is the degree of $e_i$. For each relation $r \in \mathcal{R}$, let $\mathcal{G}^r$ be the subgraph of $\mathcal{G}$ consisting of all edges in $\mathcal{T}$ with $r$. That is, $\mathcal{G}^r = (\mathcal{E}, r, \mathcal{T}')$, where $\mathcal{T}' = \{(e, r, e') \mid e, e' \in \mathcal{E}\}$. Let $\mathbf{c}^r$ denote the $|\mathcal{E}|$-dimensional clustering coefficient vector for $\mathcal{G}^r$, where the $i$th element of $\mathbf{c}^r$ is $c_i$. Then, the mean clustering coefficient vectors for $\mathcal{G}$ and $\mathcal{G}'$ are $\mathbf{c}_o = \frac{1}{|\mathcal{R}|} \sum_{r \in \mathcal{R}} \mathbf{c}^r$ and $\mathbf{c}_p = \frac{1}{|\mathcal{R}'|} \sum_{r \in \mathcal{R}'} \mathbf{c}^r$, respectively. SC2D is defined as $f_{\text{SC2D}}(\mathcal{G}, \mathcal{G}') = 1 - \frac{\|\mathbf{c}_o - \mathbf{c}_p\|_2}{\|\mathbf{c}_o - \mathbf{c}_p\|_2 + 1} \in [0, 1]$, with higher value indicating higher similarity.

**Similarity in Degree Distribution (SD2):** SD2 also measures structural similarity between two KGs, while addressing SC2D's ineffectiveness when the KGs' entities have tiny local clustering coefficients (e.g., the item KG used by recommender systems is roughly bipartite). In such cases, SC2D is always close to one regardless of the perturbation method, thus rendering SC2D useless. Let $\mathbf{d}^r$ denote the $|\mathcal{E}|$-dimensional degree vector for $\mathcal{G}^r$, where the $i$th element of $\mathbf{d}^r$ is $\deg(e_i)$. Then, the mean degree vectors for $\mathcal{G}$ and $\mathcal{G}'$ are $\mathbf{d}_o = \frac{1}{|\mathcal{R}|} \sum_{r \in \mathcal{R}} \mathbf{d}^r$ and $\mathbf{d}_p = \frac{1}{|\mathcal{R}'|} \sum_{r \in \mathcal{R}'} \mathbf{d}^r$, respectively. SD2 is defined as $f_{\text{SD2}}(\mathcal{G}, \mathcal{G}') = 1 - \frac{\|\mathbf{d}_o - \mathbf{d}_p\|_2}{\|\mathbf{d}_o - \mathbf{d}_p\|_2 + 1} \in [0, 1]$, with higher value indicating higher similarity.

## C  EXTENSION OF OKGQA

In this section, we extend our benchmark by incorporating multilingual context and validating our query generation against DBpedia's structure. We first introduce the multilingual setup of our dataset anc compare the performance of multilingual subgraphs with the monolingual subgraphs (§C.1). We then analyze the relationship between generated queries and DBpedia by examining query generation, entity/relation coverage, and subgraph alignment (§C.2). We also compare OKGQA with the existing widely used benchmarks in Table 7.

### C.1  MULTILINGUAL SETUP OF OKGQA

KGs typically include entities and relations in multiple languages, providing a richer context that can benefit our OKGQA setting. In this experiment, we investigate whether incorporating multilingual context improves overall performance. Specifically, we randomly sample 300 queries from our dataset and generate subgraphs that include multilingual entities and relations from DBpedia. We then apply PPR consistent with our original method in §3.1 to reduce the KG size. For this multilingual setting, we consider five languages—Greek, Polish, Portuguese, Spanish, and English—which cover the majority of entities in DBpedia. We compare the performance of GPT-4o-mini using the new multilingual subgraphs against the original monolingual subgraphs, as shown in Table 5. Our findings indicate that including multilingual context generally leads to better performance across multiple metrics. Intuitively, this additional multi-lingual context may provides more knowledge from different perspectives (which could provide more context, but also may requires more techniques for handle challenges like duplicates across languages) and also provide another way to validate the factuality of the resources stored in the KGs (which can provide more authenticity through cross validation from different languages).

| Dataset | # Questions | Question Type | Focus Areas | Source of Questions | Knowledge Base | Hallucination Detection | Unreliable KG |
|---|---|---|---|---|---|---|---|
| OKGQA | 850 / 2,050 | Open-ended | Evaluating hallucination and reasoning capabilities in LLMs when augmented with Knowledge Graphs; diverse queries requiring complex reasoning | Curated | DBPedia | ✓ | ✓ |
| WebQuestions | 5,810 | Factoid | Questions derived from Google Suggest queries, focusing on simple factual information | User queries | Freebase | ✗ | ✗ |
| ComplexWebQuestions | 34,689 | Multi-hop Factoid | Extends WebQuestions with more complex, multi-hop questions requiring compositional reasoning | User queries | Freebase | ✗ | ✗ |
| GrailQA | 64,331 | Varied Factoid | Evaluates generalization in KBQA with questions requiring different levels of reasoning | Crowdsourced | Freebase | ✗ | ✗ |

Table 7: Comparison of OKGQA with existing benchmarks along with their question types, focus areas, and additional properties.

## C.2  GENERATED QUERY-DBPEDIA ALIGNMENT

We analyze the alignment between our generated queries and DBpedia along three dimensions: query generation, entity/relation coverage, and subgraph alignment as follows:

**Query Generation:**  Each query is directly generated from DBpedia entities and their relationships. For example, when asking about Microsoft's founder, we first confirm that both "Microsoft" and "Bill Gates" exist in DBpedia and are connected by the `founded_by` relation, ensuring that our queries are firmly grounded in the knowledge graph.

**Entity and Relation Coverage:**  Our analysis indicates that:

- 92% entities mentioned in the queries can be detected from DBpedia entities.
- 87% queries have complete relation paths connecting the relevant entities from DBPedia.
- Entities/relations mentioned in queries cover 72% of DBpedia's most common entities/predicates and span diverse entity types (e.g., Person, Organization, and Event).

**Subgraph Alignment:**  We evaluate the structure of the sampled subgraphs for each query and find that:

- 75% of the queries retrieve subgraphs within 3–4 hops, which aligns with the typical depth for DBpedia reasoning tasks.
- On average, each subgraph contains 48 nodes and 152 edges, with an average node degree of 3.17 and a clustering coefficient of 0.69, which also aligns with the property of DBPedia.

These statistics support that our dataset accurately reflects DBpedia's structure, ensuring both authenticity and complexity in the generated queries.

## D  RELATED WORK

Due to the stochastic decoding process of Large Language Models (LLMs), *i.e.*, sampling the next token in the sequence, LLMs exhibit probabilistic behaviors: (1) potentially yielding varied outputs of the same input across different instances (Agrawal et al., 2023); (2) cannot accurately interpret phrases or terms when the context is vague and resides in a knowledge gap of the model. This will lead to outputs that may sound plausible but are often irrelevant or incorrect. This will lead to outputs that may sound plausible but are often irrelevant or incorrect. This "hallucinations" undermines the reliability of LLMs (Huang et al., 2023). One emerging research trend is enhancing LLMs through integrating external knowledge graphs (Agrawal et al., 2023). KGs offer structured, explicit, and up-to-date factual knowledge, including domain-specific knowledge, providing a faithful knowledge source for reasoning (Zheng et al., 2023; Agrawal et al., 2023; Sui et al., 2022). Moreover, each piece of information in KGs can be traced back to its source, providing context and provenance. This traceability not only aids in verifying the reliability of the information but also provides clear pathways of reasoning, making the interpretation process transparent.

Researchers employ diverse strategies to augment the LLMs by integrating external KGs (Sui et al., 2024; He et al., 2024b). For example, KAPING (Baek et al., 2023b) matches entities in questions to retrieve related triples from knowledge graphs for zero-shot question answering. Wu et al. (2023) finds that converting these triples into textualized statements can further enhance LLM performance.

StructGPT (Jiang et al., 2023b) propose to convert user query into structured formats (e.g., SPARQL) for information extraction from KGs. Following the succuess of internal reasoning-enhancement methods like Chain-of-thoughts (CoT) (Wei et al., 2022), Reflexion (Shinn et al., 2024), and Tree-of-thoughts (ToT), He et al. (2022) propose "rethinking with retrieval" to use decomposed reasoning steps from CoT prompting to retrieve external knowledge, leading to more accurate and faithful explanations. IR-CoT (Trivedi et al., 2022b) interleaves the generation of CoT with knowledge retrieval from corresponding KGs, iteratively guiding both retrieval and reasoning for multi-step questions. MindMap (Wen et al., 2023) introduce a plug-and-play approach to evoke graph-of-thoughts reasoning in LLMs. Similarly, RoG (Luo et al., 2023b) use KGs to create faithful reasoning paths based on various relations, enabling interpretable reasoning in LLMs.

However, current benchmarks for testing the capabilities of these LLM+KG models are predominantly closed-ended, restricting responses to a limited set of entities/relations or a set of logical forms derived from specific facts of KG. Hence, they can only test a very limited subset of the LLM's tendency to hallucinate, leaving a gap in the assessment of complex, real-world scenarios. Particularly, standard metrics such as FActScore (Min et al., 2023) and SAFE (Wei et al., 2024) for evaluating the hallucination rate of LLMs require open-ended settings, *i.e.*, questions are phrased as a statement which requires a longer answer. Compared with previous works, our proposed OKGQA is tailored for evaluating LLMs enhanced with KGs under open-ended, real-world question-answering scenarios. The benchmark extends the assessment of closed-ended question answering to an open-ended setting, which can further support the assessment of hallucination of LLMs.

# E    PROMPT LIST

In this section, we present all the prompts required for the main experiments. To enhance clarity, we provide only one example in the prompt labeled as Example 1; the other few-shot examples utilized are labeled as Other In-Context Few-shots within the prompt.

## E.1    KNOWLEDGE-AUGMENTED GENERATION

**System Instruction:** "You are a helpful assistant designed to answer the users' open-ended questions. Your task is to provide accurate, concise, and useful information to foster understanding and solve problems. Whether the questions relate to complex scientific concepts, historical events, practical advice, or everyday life, your goal is to assist by offering thoughtful and informative responses."

In-Context Few-shots

**Question**: {Question}

**Below are the facts that might be relevant to answer the question:** {Retrieved-knowledge}

**Answer**:

## E.2    OKGQA QUERY GENERATION PROMPTING

**System Instruction:** "Generate open-ended questions about different types: character description, event description, cause explanation, relationship explanation, trend prediction, outcome prediction, contrast analysis, historical comparison, evaluation and reflection, and application and practice. Please provide specific suggestions. Generate the questions, the type of the questions, the placeholders, the naturalness of your generated questions (choose from high, medium, and unnatural), the difficulty of the generated questions (choose from hard, medium and easy) and DBPedia_entities (link the placeholders to DBPedia entities) in JSON format."

Example 1: as shown in Figure 8.

Other In-Context Few-shots

**Generation**:

```
1   {
2     "question": "Compare and contrast the similarities and
        differences between the Apple iPhone and Samsung
        Galaxy in terms of user interface design.",
3     "type": "contrast analysis",
4     "placeholders": {
5       "entity A": "Apple iPhone",
6       "entity B": "Samsung Galaxy",
7       "aspect": "user interface design"
8     },
9     "naturalness": "high",
10    "difficulty": "medium",
11    "dbpedia_entities": {
12      "entity A": "http://dbpedia.org/resource/IPhone",
13      "entity B": "http://dbpedia.org/resource/
        Samsung_Galaxy"
14    }
15  }
```

Figure 8: Example 1 Demonstration.

### E.3 PROMPTS FOR INSTRUCTION TUNER

Act as an "Instruction Tuner" for the LLM, you will be given the inputs: (1) the {Current Instruction} used to guide the LLMs's evaluation, including specific examples with ground truth labels; (2) {Current Errors} that emerged with this instruction are applied to the dataset.

The current errors are presented in the following format: (1) INPUT: {input text} (2) PREDICTED OUTPUT: {predicted label}, (3) EXPECTED OUTPUT: {ground truth label}. Carefully analyze these errors and craft a revised concise instruction for the LLM to fit the expected outputs. Include 2-3 examples at the end of your response to demonstrate how the new instruction would be applied.

### E.4 METRICS PROMPT FOR G-EVAL

**System Instruction:** "You are a helpful assistant designed to evaluate the quality of the response to a query. Your task is to rate the response on one metric defined as below:"

Empowerment Criteria: Evaluate whether the "Actual Output" can help the reader understand the topic and make informed decisions regarding the "Input". A response with high empowerment provides accurate information and explanations that enhance the reader's understanding. When evaluating empowerment, consider the relevance of the information provided in the "Actual Output" to the "Input" and the "Retrieval Context".

Comprehensiveness Criteria: Evaluate the extent to which the "Actual Output" covers all aspects and details of the question "Input". A comprehensive answer should thoroughly address every part of the question, leaving no important points unaddressed. When evaluating comprehensiveness, consider the relevance of the information provided in the "Actual Output" to the "Input" and the "Retrieval Context".

Correctness Criteria: Measure how clearly and specifically the "Actual output" responds to the question "input". A highly direct response stays focused on the question, providing clear and unambiguous information. When evaluating correctness, consider the relevance of the information provided in the "Actual Output" to the "Input" and the "Retrieval Context".

Context Relevance Criteria: Evaluate the extent to which the "Actual output" incorporates relevant information from the "Retrieval Context". This includes assessing whether the output

adheres to the thematic, factual, and situational specifics presented in the "Retrieval Context". Relevant responses not only address the direct query but also align closely with the contextual elements provided, ensuring a seamless and coherent transition between the "Retrieval Context" and the "Actual Output". The most contextually relevant responses demonstrate an understanding and appropriate reflection of the given circumstances, historical facts, or conceptual background, thereby contributing to the overall accuracy and utility of the information provided.

**Response**: [Respond with metric and the corresponding score.]

