# OpenReview forum: "Can Knowledge Graphs Make Large Language Models More Trustworthy? An Empirical Study Over Open-ended Question Answering"
_ICLR.cc/2025/Workshop/BuildingTrust — BuildingTrust_

### Official Review · Reviewer_UP2x · 2025-03-02
**This paper proposes the OKGQA benchmark (and its variant OKGQA-P) to assess whether integrating knowledge graphs (KGs) can reduce hallucinations in large language models (LLMs) during open-ended question answering. The study introduces a KG-augmented retrieval framework—with variants based on triplets, paths, and subgraphs—and evaluates its performance using metrics such as FActScore, SAFE, and LLM-based evaluators.The paper tackles an important problem—mitigating LLM hallucinations by leveraging external structured knowledge—but falls short on several fronts.**

**Rating:** 5
**Confidence:** 4

**Review:**

Strengths

1.Relevance:
 The issue of hallucinations in LLMs is critical, and the idea of using KGs for enhanced factuality is timely.

2.Empirical Breadth:
 The authors conduct extensive experiments across multiple retrieval strategies (triplet, path, and subgraph retrieval) and evaluate various LLM backbones, which at least provides a wide empirical scope.

Weaknesses

1.Lack of Novelty:
 The proposed KG-augmented framework is a rather straightforward extension of existing retrieval-augmented generation (RAG) paradigms.

2.Methodological Shortcomings:
The benchmark's exclusive reliance on DBpedia subgraphs limits its applicability to real-world scenarios by failing to capture the diversity and dynamism of broader knowledge graphs, thereby raising significant concerns regarding the representativeness of the KG data. Moreover, the perturbation methods employed in OKGQA-P to simulate KG noise are superficial and lack rigorous motivation, further undermining the benchmark’s ability to accurately reflect real-world KG imperfections.

3.Clarity and Organization:
 The manuscript is excessively dense and suffers from poor organization. Critical details—including hyperparameter choices and the rationale behind them—are inadequately explained, making the work hard to follow.


4.Overemphasis on Empirical Comparison:
 While the empirical evaluation is extensive, the narrative lacks focus. It is unclear which aspects of KG integration truly drive improvements, and the paper does little to dissect why certain retrieval methods (e.g., subgraph retrieval) outperform others under noisy conditions.

---

### Official Review · Reviewer_m7Lk · 2025-03-02

**Rating:** 7
**Confidence:** 4

**Review:**

This work proposes OKGQA, a novel benchmark for evaluating LLM+KG on open-ended question answering tasks. By allowing models to generate paragraph-length natural responses rather than constrained outputs, it enables the use of established hallucination metrics like FActScore and SAFE. The study demonstrates that integrating knowledge graphs reduces hallucination across multiple LLM architectures, with subgraph retrieval methods performing best. The authors furthermore show that KG augmentation remains beneficial even when knowledge sources contain imperfections.

Strengths:

* This work creates the first open-ended question answering benchmark specifically designed to evaluate hallucination in KG-augmented LLMs.
* It tests multiple LLM architectures (GPT-4o, Llama-3.1, Mistral, Gemma) and various KG integration methods.
* It employs multiple evaluation metrics (FActScore, SAFE, G-Eval) with validation through human correlation studies
* The authors investigated robustness when using imperfect knowledge sources through the OKGQA-P variant, and found the KG still improves the performance.

Weaknesses:
* The dataset relies exclusively on DBpedia, limiting generalizability to other knowledge graphs.
* Data in the datasets is generated by templates, which may not perfectly represent real-world KG usage.
* Limited exploration of how the size and structure of retrieved knowledge affects performance.
* Only evaluated one graph retrieval method from the literature and would be beneficial to add more evaluated methods.

---

### Decision · Program_Chairs · 2025-03-04

Accept